# Study on crude oil displacement efficiency by fracturing fluid in tight sandstone reservoir

**Jinfeng Li[1], Xingjun Gao[1], Lianru Yang[1], Jiashun Gong[1], Tezheng Duan[1], Fan Song[1], Jinsheng Zhao[2]\***

**1** Xiasiwan Oil Production Plant, Yanchang Oilfield Co., Ltd., Yan'an, Shaanxi, China, **2** Institute of Carbon Neutrality Future Technology, Xi'an Shiyou University, Xi'an, Shaanxi, China

\* jingsheng79317@163.com

## Abstract

Tight sandstone reservoirs usually experience a long flowback period after hydraulic fracturing, which significantly affects oil production. After fracturing, the well-soaking is commonly employed to control fracturing fluid flowback and enhance oil recovery, so that the oil in the reservoir matrix is replaced by fracturing fluid, which can improve the crude oil recovery and reduce the flowback of the fracturing fluid. In this paper, the gel breaking fluid of slick water fracturing fluid, guanidine gum fracturing fluid and cross-linked guanidine gum fracturing fluid are used as displacement working fluids to study the effect of gel breaking fluid on oil displacement in tight sandstone reservoirs. The results show that it is not the smaller the pore radius that the higher the displacement efficiency, but the oil displacement efficiency is higher for the equilibrium of capillary force and percolation resistance in a certain radius of pore throat. For tight sandstone reservoir, the oil displacement efficiency of small pore, middle pore and large pore is higher, the oil displacement efficiency of micro-pore is lower, and the oil in pore throat with radius less than 0.01 μm is difficult to be replaced. The lower the interfacial tension is, the lower the viscosity is, and the higher the oil displacement efficiency is. For tight sandstone reservoir, the displacement efficiency of crude oil can reach 13.11% −33.31%, the displacement of crude oil in the early stage is mainly displaced out of the middle and small pores, and then replaced out of the large pores and micro-pores of crude oil.

## 1. Introduction

Tight sandstone reservoirs are characterized by low permeability, nanoscale pore-throat development, and poor reservoir properties, with significant geological complexity and engineering challenges. At present, research on enhanced oil recovery (EOR) in tight sandstone reservoirs has entered a new stage of "physical–chemical–gas flooding" synergistic development. Techniques such as imbibition after fracturing and $CO_2$ injection (including both flooding and huff-n-puff methods) have

**Data availability statement:** All relevant data are within the manuscript and its Supporting Information files.

**Funding:** This research is supported by the National Natural Science Foundation of China (No.52174031) and the National Natural Science Foundation of China (No.52174031), and the Youth Innovation Team of Shaanxi Universities.

**Competing interests:** The authors have declared that no competing interests exist.

been widely applied in practice [1]. And tight reservoirs are usually accompanied by a relatively long flowback period after fracturing operations. To maximize the utilization of flowback fluid during this stage, enhance formation energy, and effectively produce crude oil in the near-wellbore area, a special production method—well-soaking and controlled flowback technique—is commonly adopted after fracturing. This method allows for thorough oil-water displacement between the flowback fluid in propped fractures and the oil in reservoir matrix. During the soaking stage, the fracturing fluid spontaneously enters into the rock pores through the imbibition effect, displacing the crude oil within them. This process usually occurs between the fractures created by fracturing and the reservoir matrix. After the soaking stage, the displaced crude oil is produced together with the fracturing fluid during the flowback stage, forming an oil-water mixture that significantly increases crude oil production and reduces the flowback volume of fracturing fluid [2–6].

The main mechanism of oil-water displacement is imbibition oil recovery, which has been widely studied in tight oil reservoirs in recent years. Imbibition is more common in low-permeability and tight reservoirs due to the presence of microscopic pore throats and fractures formed during hydraulic fracturing. Since the 1950s, imbibition has been extensively studied and developed as an important method of oil production [7]. Spontaneous imbibition refers to the process in which the wetting phase displaces the non-wetting phase fluid from the porous medium under the combined action of capillary force and gravity [8–10]. Imbibition is a complex process affected by various factors, such as reservoir pore characteristics, oil-water interfacial tension, wettability, and formation fluid properties. Based on previous studies, Yang et al. concluded that pore structure factors, such as pore size, pore-throat type, and connectivity, affect capillary force and thus spontaneous imbibition efficiency [11]. Cuiec, Morrow, and others found that lower interfacial tension reduces imbibition efficiency [12,13]. Li et al. demonstrated that surfactants can alter the wettability of water-wet surfaces, reduce capillary pressure, and improve imbibition recovery [14]. Austad et al. studied and showed that the addition of surfactants can change rock wettability, thereby improving imbibition efficiency [15,16]. Considering the widely used guanidine gum fracturing fluid system, its gel-breaking flowback fluid often contains abundant surfactants, exhibiting excellent interfacial activity. Previous studies have shown that, in addition to the superior properties of the reservoir itself, adding surfactants, reducing the viscosity of the displacement fluid, and increasing the displacement time can effectively improve the oil displacement efficiency in tight oil reservoirs [17–19]. In addition, Zhu et al. conducted core imbibition experiments under different crude oil viscosities and core permeabilities, and the results showed that imbibition recovery increased with the decreasing oil viscosity and the increasing core permeability [7].

In summary, previous studies mainly focused on the influence of reservoir properties and formation fluid properties on oil-water displacement, while the displacement efficiency between different fracturing fluids and crude oil during the fracturing fluid flowback process has been less studied, and the effects of fracturing fluid properties such as fracturing fluid type and viscosity on displacement efficiency have also been

rarely investigated. Therefore, this paper selects the gel-breaking fluids of slick water fracturing fluid, guanidine gum fracturing fluid, and linear gel fracturing fluid as displacement fluids to conduct an in-depth study on the oil displacement effect of these gel-breaking fluids in tight oil reservoirs. Meanwhile, advanced nuclear magnetic resonance (NMR) technology is employed to investigate the pore-scale oil mobilization characteristics of core samples under different experimental conditions, clarifying the oil recovery behavior within various types of pores under different influencing factors and elucidating the mechanisms by which different fracturing fluid systems displace crude oil. These research results will provide a strong theoretical and practical reference for optimizing the post-fracturing flowback system and guiding the selection of fracturing fluid types in tight oil reservoirs.

## 2. Experimental section

### 2.1 Experimental materials and scheme

Five numbered cores were used in the experiment. The experimental oil was simulated reservoir crude oil with a viscosity of 2.2 mPa·s at 80°C. The displacement fluids included simulated formation water, fracturing fluid gel-breaking fluids, and surfactant solutions. The core physical properties, displacement fluid properties, and experimental conditions are shown in Table 1.

### 2.2. Experimental procedures

According to the experimental scheme, the following procedures were developed, and the experimental process is shown in Fig 1.

(1) After testing the porosity and permeability of the natural sandstone cores, the cores were dried, vacuumed, and then saturated under high pressure with formation water containing 25000 mg/L of $MnCl_2$.

(2) Inject simulated crude oil into the core saturated with formation water until the fluid at the core outlet contains 100% oil.

(3) Conduct NMR testing on the oil-saturated core to obtain the original oil distribution.

(4) Under simulated reservoir temperature conditions, the prepared cores were immersed in different types of fracturing fluids. According to the experimental design shown in Table 1, the type of working fluid, the proportion of additives, and the soaking time were varied. The overall experimental procedure is illustrated in Fig 1.

(5) After soaking for a certain time, perform NMR testing on the core again to obtain the oil distribution after oil-water displacement.

(6) Analyze the displacement efficiency in various pores based on the $T_2$ distribution curves before and after oil-water displacement.

**Table 1. Experimental material properties and experimental scheme.**

| Core number | Length (cm) | Diameter (cm) | Porosity (%) | Permeability ($10^{-3}\mu m^2$) | Types of displacement fluid | Oil-water interfacial tension (mN/m) | Displacement time (h) |
|---|---|---|---|---|---|---|---|
| 1 | 3.400 | 2.520 | 5.83 | 0.121 | Guanidine gum gel-breaking fluid | 0.059 | 24 |
| 2 | 3.400 | 2.520 | 5.83 | 0.121 | Cross-linked guanidine gum breaking fluid | 0.065 | 24 |
| 3 | 2.615 | 2.520 | 8.62 | 0.110 | Slick water breaking fluid | 0.613 | 24 |
| 4 | 2.615 | 2.520 | 8.62 | 0.110 | Slick water breaking fluid+0.3%OC-1 | 0.055 | 24 |
| 5 | 4.980 | 2.520 | 7.31 | 0.149 | Guanidine gum gel-breaking fluid | 0.059 | 4,8,12,16 |

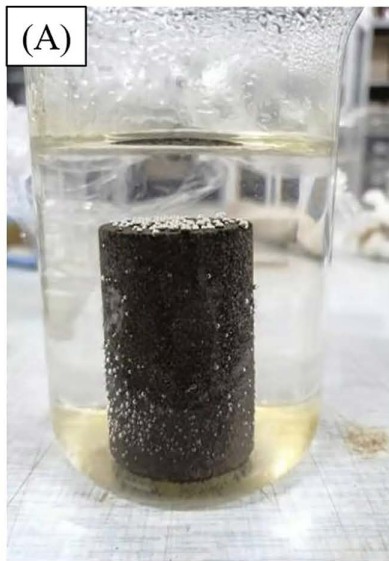
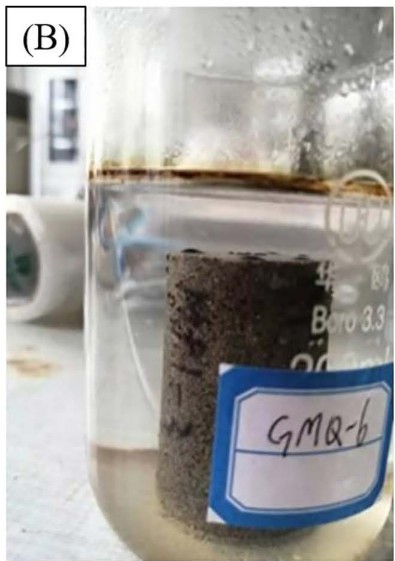
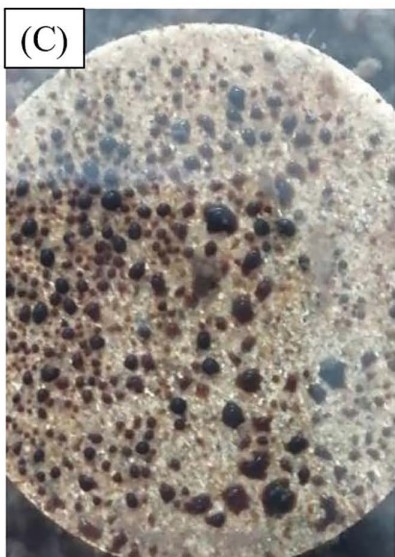

**Fig 1. Oil-water displacement experiment diagram. (A)** Early stage of oil-water displacement. **(B)** Middle stage of oil-water displacement. **(C)** Late stage of oil-water displacement.

### 2.3. Calculation of crude oil displacement efficiency

In this study, a MesoMR23–060 NMR imaging system (Suzhou Niumag Analytical Instrument Corporation, Suzhou, China) was used for core testing to investigate the distribution of crude oil within the core samples. This equipment is equipped with an independent pulse control module and a radio frequency (RF) transmission/reception circuit, which improves the accuracy and stability of pulse sequences and enhances the signal-to-noise ratio of NMR signals. The spectrometer frequency of the RF pulse used in this study was 21 MHz, and the waiting time between two scans of sampling data was 4000 ms.

The $T_2$ spectrum (transverse relaxation time spectrum) is an important parameter in NMR logging, reflecting the relaxation characteristics of fluids in rock pores. Generally, larger pores have longer $T_2$ values, while smaller pores (such as micropores) have shorter $T_2$ values [20,21]. The signal amplitude on the vertical axis represents the crude oil content within pores of the corresponding size, and a decrease in signal amplitude after the experiment indicates that crude oil in those pores has been displaced and produced. When oil-water displacement occurs, oil in the pores is replaced by water. Due to the different relaxation characteristics of water and oil, this leads to changes in the $T_2$ spectrum. Especially when oil in large and medium pores is replaced by water, the $T_2$ spectrum curve will significantly decrease in these regions due to the significant reduction in oil content [22]. Because tight cores have fine pore-throats and very low pore volume, the amount of oil displaced during the oil-water displacement process is minimal. Conventional manual measurement methods have large errors. To address this issue, this experiment calculates the displacement efficiency using the NMR $T_2$ differences before and after oil-water displacement [23], with the following formula:

$$R = \frac{S_O - S}{S_O}$$

(1)

Where $R$ is the crude oil displacement efficiency, %; $S$ is the area enclosed by the NMR $T_2$ spectrum and the $T_2$ axis after oil-water displacement; $S_O$ is the area enclosed by the NMR $T_2$ spectrum and the $T_2$ axis when the core is saturated with oil.

## 2.4. Conversion relationship between NMR $T_2$ value and pore-throat radius

According to the common classification method of $T_2$ spectra in tight reservoirs, pore-throat sizes are divided into four categories [23]: micropores ($T_2 < 1$ ms), small pores ($1$ ms $< T_2 < 10$ ms), medium pores ($10$ ms $< T_2 < 100$ ms), and large pores ($T_2 > 100$ ms). To more intuitively represent the pore-throat size distribution of each core sample, the abscissa of the NMR $T_2$ spectrum (transverse relaxation time $T_2$, ms) needs to be converted to pore-throat radius r (μm). The conversion of NMR $T_2$ relaxation time to pore-throat radius is a widely accepted approach for pore structure characterization in tight reservoirs [24,25]. And this can be achieved by correlating the pore-throat radius distribution curve from mercury injection testing with the $T_2$ spectrum distribution curve from NMR testing for the core sample to find the peak-corresponding pore-throat radius and $T_2$, thereby calculating the conversion coefficient C using the formula:($r = C \times T_2$). As shown in Figs 2 and 3, they are the pore-throat radius distribution curve from mercury injection testing and the $T_2$ spectrum distribution curve from NMR testing for the core sample, respectively. By matching the first peaks of the two graphs, the conversion coefficient $C_1 = 0.016/0.3 = 0.053$ μm/ms is obtained. By matching the second peaks, $C_2 = 0.25/8.4 = 0.03$ μm/ms is obtained. The average of the two coefficients is $C = (C_1 + C_2)/2 = 0.04$. Thus, the four categories of pore-throats represented by $T_2$ values

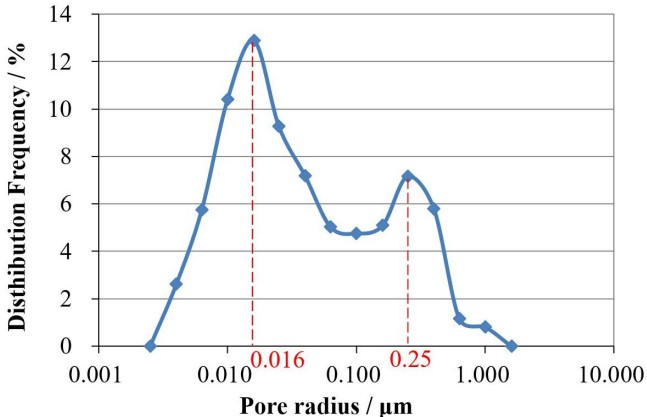

**Fig 2. Pore throat radius distribution of rock sample by mercury injection test.**

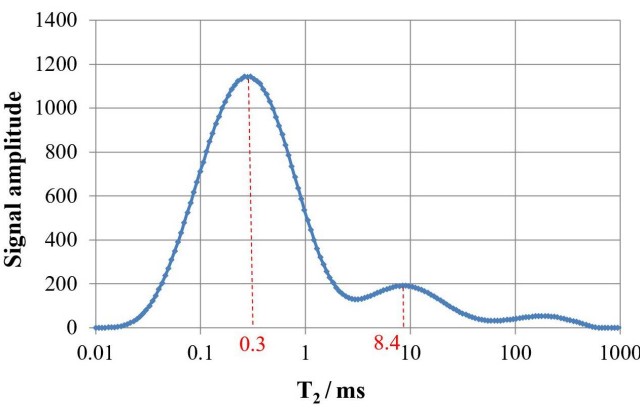

**Fig 3. $T_2$ spectrum distribution of rock samples by nuclear magnetic resonance test.**

can be expressed in terms of pore-throat radius: micropores (r < 0.04 μm), small pores (0.04 μm < r < 0.4 μm), medium pores (0.4 μm < r < 4 μm), and large pores (r > 4 μm).

## 3. Experimental results and analysis

### 3.1. Influence of displacement fluid viscosity on crude oil displacement efficiency

To investigate the influence of displacement fluid viscosity on displacement efficiency, a long core was cut into two short cores, ensuring identical permeability. The displacement fluids used were guanidine gum gel-breaking fluid and cross-linked guanidine gum gel-breaking fluid, with close interfacial tensions of 0.059 mN/m and 0.065 mN/m, respectively. The viscosities of the two gel-breaking fluids at 80°C were 2.1 mPa·s and 4.3 mPa·s, respectively. The $T_2$ spectrum distributions before and after oil-water displacement for the two core samples are shown in Fig 4, and the displacement efficiencies in various pores are listed in Table 2.

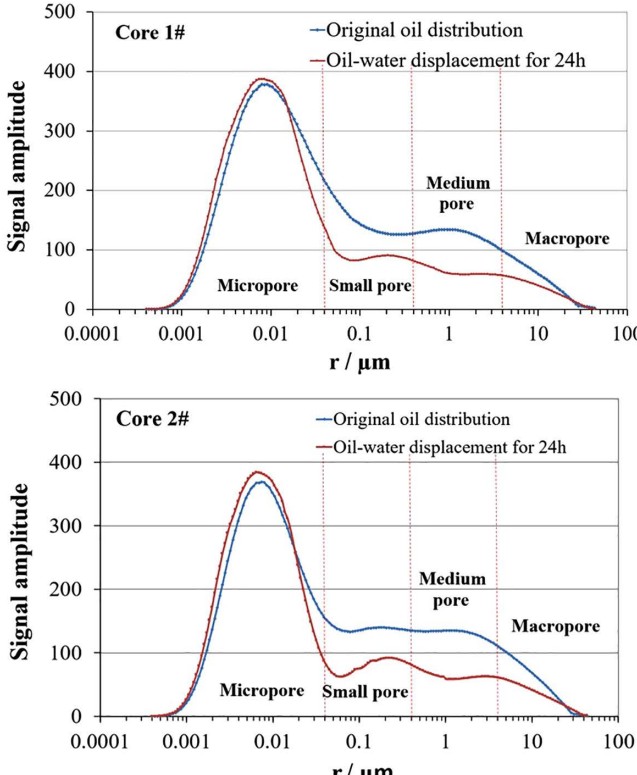

**Fig 4. The oil distribution curves of cores 1# and 2# before and after oil-water displacement.**

**Table 2. The oil displacement efficiency of cores 1# and 2#.**

| Core number | Displacement efficiency (%) | | | | |
|---|---|---|---|---|---|
| | Micropore (<0.04μm) | Small pore (0.04–0.4μm) | Medium pore (<0.4–4μm) | Large pore (>4μm) | Synthesis |
| 1 | −0.94 | 38.63 | 80.81 | 33.14 | 23.25 |
| 2 | −4.17 | 42.44 | 50.98 | 37.06 | 18.31 |

As shown in Fig 4 and Table 2, since the two core samples were cut from the same long core, they have identical permeability and similar pore-throat distributions. The NMR results clearly show that after 24 hours of oil-water displacement, the signal amplitude of the cores is significantly lower than that of the oil-saturated cores, particularly in the regions corresponding to small and medium pores. This indicates that crude oil in these pores is preferentially displaced, leading to a marked reduction in oil content. And both samples exhibit the highest crude oil displacement efficiency in medium pores, followed by small and large pores, while the crude oil in micropores increased, resulting in negative displacement efficiency. The overall crude oil displacement efficiency of core 1 is 4.94% higher than that of core 2.

Analysis shows that when the oil-water interfacial tensions of the displacement fluids used in the two core samples are close, the difference in displacement efficiency is mainly caused by the viscosity difference of the displacement fluids. High-viscosity displacement fluids encounter greater flow resistance in rock pores. Due to the small geometric size of these pores, high-viscosity fluids cannot effectively enter and displace the crude oil. And low-viscosity fluids exhibit better mobility, enabling them to enter small and medium pores more effectively, establish stable flow pathways, and promote efficient displacement. According to the experimental results, the higher the viscosity of the displacement fluid, the lower the crude oil displacement efficiency. The phenomenon that the oil in micropores not only did not decrease but increased, leading to the oil-water displacement efficiency in core micropores being less than zero, is analyzed to be due to the migration of crude oil from small, medium, and large pores to micropores during the oil-water displacement process, causing an increase in crude oil in some micropores.

Therefore, viscosity is a key controlling factor for displacement efficiency. Lower-viscosity fracturing fluids achieve higher oil recovery by enhancing displacement in small and medium pores, whereas higher-viscosity fluids are less effective and may even contribute to oil redistribution into micropores. Hence, in the process of fracturing, special attention should be paid to the controlling effect of displacement fluid viscosity on oil recovery performance.

## 3.2. Influence of surfactants on crude oil displacement efficiency

Aiming at the situation that the gel-breaking fluid of slick water fracturing fluid has no flowback aid component, resulting in high interfacial tension and low crude oil displacement efficiency. Therefore, two cores 3 and 4 with exactly the same permeability were selected, and the displacement fluids were respectively the gel-breaking fluid of slick water fracturing fluid and the gel-breaking fluid of slick water fracturing fluid added with surfactant, with oil-water interfacial tensions of 0.613mN/m and 0.055mN/m, and the same viscosity of the two displacement fluids, so as to investigate the influence of surfactant on the displacement efficiency of the gel-breaking fluid of slick water fracturing fluid. The oil distribution before and after oil-water displacement of the two core samples is shown in Fig 5, and the crude oil displacement efficiency in various pores is shown in Table 3.

As shown in Fig 5 and Table 3, since the two core samples were cut from the same long core, they have the same permeability and similar original oil distribution. In the four types of pores, the crude oil displacement efficiency in small pores and medium pores is the highest, followed by large pores. The crude oil displacement efficiency in micropores, like that in the previous core samples, shows the phenomenon that the crude oil does not decrease but increase. For tight sandstone, due to the too small pore-throat size of micropores, although the capillary force is large, the seepage resistance is also large, resulting in poor displacement effect. For large pores, the pore-throat size is large, and although the seepage resistance of oil and water in them is small, the capillary force as the driving force of oil-water displacement is too small to achieve higher crude oil displacement efficiency. It can be seen that only within a certain range of pore-throat size, when the capillary force and seepage resistance reach an equilibrium state, a higher crude oil displacement efficiency can be formed.

On the whole, because the displacement fluid of core 4 is added with surfactant, which makes the interfacial tension of the gel-breaking fluid lower, the displacement efficiency is 3.62% higher than that of the gel-breaking fluid of slick water fracturing fluid without surfactant. The addition of surfactants can greatly reduce the oil-water interfacial tension, which

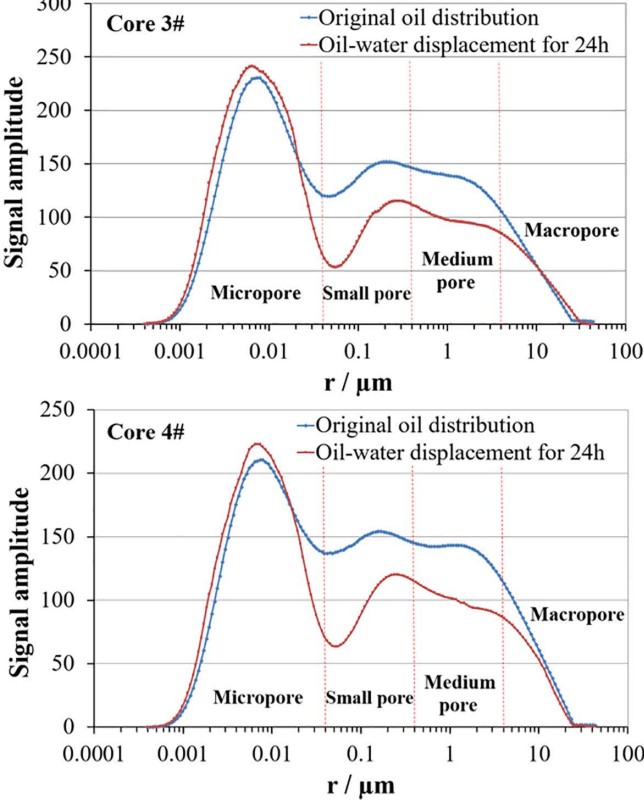

**Fig 5. The oil distribution curves of cores 3# and 4# before and after oil-water displacement.**

**Table 3. The oil displacement efficiency of cores 3 and 4.**

| Core number | Displacement efficiency (%) | | | | |
|---|---|---|---|---|---|
| | Micropore (<0.04μm) | Small pore (0.04–0.4μm) | Medium pore (<0.4–4μm) | Large pore (>4μm) | Synthesis |
| 3 | −6.51 | 36.61 | 27.37 | 5.31 | 13.11 |
| 4 | −1.63 | 34.62 | 27.98 | 19.08 | 16.73 |

enhances the deformability of crude oil droplets, allowing them to flow smoothly through pore throats where they would otherwise be trapped under high interfacial tension conditions during the imbibition process [7]. This promotes better crude oil mobility, enlarges the swept volume, and improves displacement efficiency. In addition, a lower interfacial tension can also reduce the adhesion work of crude oil, facilitating its detachment from the rock surface. And with the decreasing adhesion work, the oil-washing efficiency increases. Therefore, it can be concluded that surfactants can reduce the interfacial tension of the displacement fluid, weaken the interaction between oil and water, and thus improve the displacement efficiency.

### 3.3. Influence of displacement time on crude oil displacement efficiency

To investigate the influence of displacement time on crude oil displacement efficiency, the same oil-saturated core was immersed in fracturing fluid for different durations. The effect of displacement time on micro-displacement efficiency was analyzed based on the $T_2$ spectrum curves measured after different displacement times. After converting $T_2$ values into

pore-throat radii, the crude oil distribution in the core at different displacement times is shown in Fig 6, and the displacement efficiencies in various pore types at different displacement times are listed in Table 4.

As shown in Fig 6 and Table 4, with the increase of displacement time, the overall oil distribution curve shows a downward trend, and the crude oil displacement efficiency gradually improves. In the early stage of displacement (0–4 hours), the displaced crude oil mainly comes from small and medium pores, with displacement efficiencies of 33.51% and 38.17%, respectively. The displacement efficiency in large pores is only 9.27%, and the oil volume in micropores slightly increases, with an overall efficiency of 11.17%. During the 4–8 hours stage, the displacement efficiencies of large pores and micropores increased the most, by 6.61% and 3.68%, respectively, while those of small and medium pores increased less, with an overall efficiency increase of 3.35% to 14.52%. During the 8–12 hours period, the overall displacement efficiency increased the most, with the displacement efficiency of large pores increasing by 31.88%, that of micropores by 13.15%, and those of small and medium pores increasing less, with an overall efficiency increase of 13.15% to 27.67%. In the 12–16 hours stage, except for medium pores, the growth rates of displacement efficiencies in other pores slowed down, all less than 10%, and the overall efficiency increased by 5.64% to 33.31%. The experiment showed that when the displacement time was extended to 20 hours, no more crude oil was displaced from the core. Finally, the displacement efficiency of large pores was the highest at 56.18%, followed by medium and small pores (49.45%, 49.33%), and the lowest in micropores at 23.45%. It can be seen that in the early stage of oil-water displacement in tight sandstone, small and medium pores are the main contributors (capillary force and seepage resistance reach a relative equilibrium state, forming higher crude oil displacement efficiency in the early stage). As the crude oil in medium and small pores decreases, the capillary force, the driving force for overall imbibition, weakens, and the displacement of crude oil in large pores becomes dominant, followed by micropores. Eventually, the displaced crude oil is mainly from small, medium, and large pores.

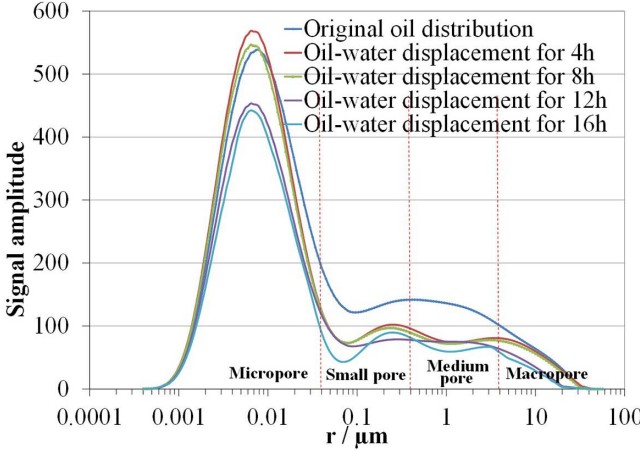

**Fig 6. The oil distribution curves of core 5 at different displacement time.**

**Table 4. The oil displacement efficiency in various pores of core 5# with different displacement time.**

| Displacement time(h) | Displacement efficiency (%) | | | | |
|---|---|---|---|---|---|
| | Micropore (<0.04μm) | Small pore (0.04–0.4μm) | Medium pore (<0.4–4μm) | Large pore (>4μm) | Synthesis |
| 4 | −0.47 | 33.51 | 38.17 | 9.27 | 11.17 |
| 8 | 3.21 | 35.14 | 40.56 | 15.88 | 14.52 |
| 12 | 18.34 | 43.26 | 43.06 | 47.76 | 27.67 |
| 16 | 23.45 | 49.33 | 49.45 | 56.18 | 33.31 |

## 4. Conclusions

In this study, different types of fracturing fluids are selected to conduct oil-water displacement experiments on tight sandstone cores. Combined with NMR testing, the pore-scale oil mobilization characteristics are clarified, and the effects of surfactant addition and displacement conditions on displacement efficiency are identified. These findings provide theoretical and practical guidance for optimizing post-fracturing flowback systems and selecting appropriate fracturing fluid types in tight oil reservoirs. The following conclusions can be drawn from this study:

(1) During the displacement process between fracturing fluid and oil after fracturing, a smaller pore radius does not necessarily result in higher displacement efficiency. Within a specific pore-throat radius range, where capillary force and seepage resistance reach equilibrium, the crude oil displacement efficiency is higher. For tight sandstone, the crude oil displacement efficiency is higher in small pores, medium pores, and large pores, while lower in micropores.

(2) The displacement efficiency of the low-viscosity gel-breaking fluid is 4.94% higher than that of the high-viscosity fluid. Low-viscosity fluids exhibit better mobility, enabling them to enter small and medium pores more effectively, establish stable flow pathways, and promote efficient displacement.

(3) The displacement efficiency of the gel-breaking fluid containing surfactant is 3.62% higher than that of the fluid without surfactant. By significantly reducing oil-water interfacial tension, the addition of surfactants enhances the deformability and mobility of crude oil droplets, thereby enlarging the swept volume and improving displacement efficiency.

(4) For tight sandstone under the crude oil conditions used in this experiment, the crude oil displacement efficiency can reach 13.11%−33.31%. In the early stage of displacement, crude oil is mainly displaced from medium and small pores, followed by large pores and micropores in subsequent stages. Crude oil in pore-throats with a radius smaller than 0.01 μm is basically difficult to displace.

## Supporting information

**S1 File. Method for Converting NMR $T_2$ Values to Pore Radius.** The fundamental principle and detailed derivation process for converting the NMR relaxation time into pore size of core samples through high-pressure mercury intrusion experiments and nuclear magnetic resonance (NMR) testing.
(PDF)

**S1 Table. NMR data.** The specific NMR data of the core samples after oil saturation and after the experiment.
(XLSX)

## Author contributions

**Data curation:** Jinfeng Li, Lianru Yang, Fan Song.

**Formal analysis:** Jinfeng Li, Jiashun Gong, Tezheng Duan.

**Investigation:** Jinfeng Li, Lianru Yang, Jiashun Gong, jinsheng zhao.

**Methodology:** Jinfeng Li, Lianru Yang, Tezheng Duan.

**Resources:** Lianru Yang.

**Supervision:** Xingjun Gao.

**Writing – original draft:** jinsheng zhao.

**Writing – review & editing:** Xingjun Gao, jinsheng zhao.

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
