## [Decision Letter · Decision Letter 0]

3 Aug 2025

Dear Dr. zhao,

Thank you for submitting your manuscript to PLOS ONE. After careful consideration, we feel that it has merit but does not fully meet PLOS ONE’s publication criteria as it currently stands. Therefore, we invite you to submit a revised version of the manuscript that addresses the points raised during the review process.

We look forward to receiving your revised manuscript.

Kind regards,

Lisong Zhang

Academic Editor

PLOS ONE

Journal Requirements:

2. We note that this submission includes NMR spectroscopy data. We would recommend that you include the following information in your methods section or as Supporting Information files:

1) The make/source of the NMR instrument used in your study, as well as the magnetic field strength. For each individual experiment, please also list: the nucleus being measured; the sample concentration; the solvent in which the sample is dissolved and if solvent signal suppression was used; the reference standard and the temperature.

2) A list of the chemical shifts for all compounds characterised by NMR spectroscopy, specifying, where relevant: the chemical shift (δ), the multiplicity and the coupling constants (in Hz), for the appropriate nuclei used for assignment.

3)The full integrated NMR spectrum, clearly labelled with the compound name and chemical structure.

We also strongly encourage authors to provide primary NMR data files, in particular for new compounds which have not been characterised in the existing literature. Authors should provide the acquisition data, FID files and processing parameters for each experiment, clearly labelled with the compound name and identifier, as well as a structure file for each provided dataset. See our list of recommended repositories here: https://journals.plos.org/plosone/s/recommended-repositories

This research is supported by the National Natural Science Foundation of China No.52174031�and the Youth Innovation Team of Shaanxi Universities.

This research is supported by the National Natural Science Foundation of China No.52174031�and the Youth Innovation Team of Shaanxi Universities.

This research is supported by the National Natural Science Foundation of China No.52174031�and the Youth Innovation Team of Shaanxi Universities.

6. Please amend your authorship list in your manuscript file to include author jinsheng zhao.

7. Please amend the manuscript submission data (via Edit Submission) to include author Li JInfeng.

Additional Editor Comments :

Reviewer 1

This manuscript focuses on the study of the displacement efficiency of fracturing fluid on crude oil in tight sandstone reservoirs, conducting relevant experiments, and exhibiting certain innovativeness and practicality. However, there are some deficiencies need to be checked out and revised:

1. In Introduction section, author mentioned imbibition, but there is a lack of the relevance of imbibition to the main work of this manuscript. It is recommended that author should clarify this relevance to enhanced the readability of the manuscript.

2. The relationship of imbibition and the flowback of fracturingfluid should be clarified in introduction (for example, the flowback of fracturingfluid may be achieved through imbibition).

3. The case of letters in table 1 needs to be consistent.

4. It is recommended that other physical parameters of experimental cores such as length, diameter and porosity should be shown in table 1.

5. Is the method that converting theabscissa of the NMR T₂ spectrum to pore-throat radius commonly used in the industry? Author should appropriately cite some relevant literatures to demonstrate its rationality.

6. Introduction section mentioned that the study revealed the main factors affecting oil-water displacement during the flowback process of tightoil fracturing, but the discussions of main factorsin Section 2 and 3 is not enough, which is confused.

7. The manuscript only reviewed the effect of capillary force on imbibition, while gravity also plays an important role in affecting imbibition. To better explain the background and relevant theoriesof this study, it is recommended to cite the following literatures:

a) https://doi.org/10.1016/j.geoen.2024.213071.

b) https://doi.org/10.2118/224403-PA.

c) https://doi.org/10.1080/01932691.2023.2177670.

d) https://doi.org/10.1016/j.geoen.2024.213057.

Reviewer 2

The paper is lacking with the several important results. The present form of the manuscript is not acceptable for possible publication in the journal. The comments that need to be addressed as follows:

1. While the authors have applied the tight sandstone method in their study, it would be beneficial to provide a brief introduction to this method for a clearer understanding by readers. Very few manuscripts have been considered to address the issues which is not sufficient.

2. The inclusion of Table 1, comparing results across various studies, is appreciated. However, the novelty of the study could be better emphasized. Strengthening the content related to the novelty would enhance the paper.

3. Although the introduction section offers a comprehensive overview, it would be beneficial to include more recent works in the field to further contextualize the study.

4. Full methodology section is incomplete such as sample preparation, testing and error analysis.

5. Authors could add more explanation for this statement. 'Fig. 4 - Fig. 6 were sketched to portrayed the streamline graphs for the signal amplitude problems under discussion for distinct values of pores. They stated that increasing in the pores, the graphs of streamline more presiding.'

6. Many of the study's results are presented without thorough explanation. Adding citations in the results and discussion section would help properly convey the significance of each finding and provide a more comprehensive analysis.

7. Several minor typographical errors are present in the paper. Correcting these errors is essential for improved readability and credibility.

8. The axes labels and tick labels in the figures are too small, affecting clarity. Enhancing the font sizes of these labels would significantly improve readability.

9. The conclusion section is not thoroughly presented.

Reviewer 3

Hello, Author. Thank you for submitting this manuscript for review. I want to commend the thoroughness of the research; however, I have highlighted its strengths and some areas where it could be improved.

1. This study addresses relevant practical problems like long flowback in tight reservoirs. It employs suitable experimental methods such as coreflooding and NMR T2 analysis to examine the displacement efficiency of different fracturing fluid gel-breakers. The experimental design controls key variables such as permeability through core pairs, interfacial tension, and viscosity to isolate their effects. Using NMR for pore-level displacement measurement is a strength. The conclusions about pore size impact, viscosity, interfacial tension, and time dependency are reasonable extensions of the experimental data.

Tables 2, 3, and 4 and Figures 4, 5, and 6 generally support the main conclusions regarding the effects of viscosity (Table 2), surfactants/interfacial tension (Table 3), and displacement time (Table 4) on overall and pore-specific displacement efficiency. The data support the observation of negative efficiency in micropores and plausibly explained.

However, the main weakness is the lack of replication. The study uses only one core per condition (e.g., one core for guanidine gum, one for cross-linked gum, one for slickwater, one for slickwater+surfactant). Although cores from longer samples were used for some comparisons (cores 1-2, 3-4), this does not constitute true replication. Natural heterogeneity within a single long core means results from one sample may not be representative. Without replicates, the reliability and general applicability of the efficiency values (e.g., 23.25% vs 18.31%, 13.11% vs 16.73%) are significantly reduced. While the directions of effects (e.g., lower viscosity increases efficiency) are probably valid due to controlled comparisons, the effect magnitudes and absolute efficiency values lack solid statistical support since n=1 per condition. The time-dependent study (Core 5) also involved only one core sequentially.

2. The manuscript reports no statistical analysis at all. Results are presented as single measurements (e.g., displacement efficiencies in tables) without any variance measures (standard deviation, error bars), significance tests (t-tests, ANOVA), or discussion of uncertainty. Given that only one core per condition was used (n=1), rigorous statistical comparison of treatments is impossible. The lack of acknowledgment of this limitation or attempts to quantify uncertainty (such as repeated measurements if feasible) means the analysis is not rigorous or appropriate.

3. The manuscript does not specify that raw data (like NMR T2 spectra, core properties beyond permeability, fluid measurements) have been deposited in a public repository or provided as supplementary material. While summary results are shown in figures and tables, raw data necessary for reproduction or re-analysis are not explicitly available in the manuscript or declared as accessible elsewhere.

4. The manuscript is generally clear and follows the conventional structure (Abstract, Intro, Methods, Results, Discussion, Conclusion). Technical terminology is appropriate. However, there are some instances of non-standard English and grammatical errors that slightly hinder readability but do not obscure the meaning:

a. "There usually have a long backflow period" (Abstract - should be "is").

b. "produced by the well soak" (Abstract - phrasing is awkward).

c. "reservoir reservoirs" (Abstract - redundant)

d. Inconsistent capitalization (e.g., "Guanidine gum" vs "guanidine gum").

e. Sentences are often lengthy and would benefit from concise rewriting and professional copyediting.

Reviewers' comments:

Reviewer's Responses to Questions

**Comments to the Author**

1. Is the manuscript technically sound, and do the data support the conclusions?

Reviewer #1: Yes

Reviewer #2: No

Reviewer #3: Partly

2. Has the statistical analysis been performed appropriately and rigorously?

Reviewer #1: Yes

Reviewer #2: No

Reviewer #3: No

3. Have the authors made all data underlying the findings in their manuscript fully available?

Reviewer #1: Yes

Reviewer #2: No

Reviewer #3: No

4. Is the manuscript presented in an intelligible fashion and written in standard English?

Reviewer #1: Yes

Reviewer #2: No

Reviewer #3: No

Reviewer #1: This manuscript focuses on the study of the displacement efficiency of fracturing fluid on crude oil in tight sandstone reservoirs, conducting relevant experiments, and exhibiting certain innovativeness and practicality. However, there are some deficiencies need to be checked out and revised.

Reviewer #2: The paper is lacking with the several important results. The present form of the manuscript is not acceptable for possible publication in the journal. The comments that need to be addressed as follows:

1. While the authors have applied the tight sandstone method in their study, it would be beneficial to provide a brief introduction to this method for a clearer understanding by readers. Very few manuscripts have been considered to address the issues which is not sufficient.

2. The inclusion of Table 1, comparing results across various studies, is appreciated. However, the novelty of the study could be better emphasized. Strengthening the content related to the novelty would enhance the paper.

3. Although the introduction section offers a comprehensive overview, it would be beneficial to include more recent works in the field to further contextualize the study.

4. Full methodology section is incomplete such as sample preparation, testing and error analysis.

5. Authors could add more explanation for this statement. 'Fig. 4 - Fig. 6 were sketched to portrayed the streamline graphs for the signal amplitude problems under discussion for distinct values of pores. They stated that increasing in the pores, the graphs of streamline more presiding.'

6. Many of the study's results are presented without thorough explanation. Adding citations in the results and discussion section would help properly convey the significance of each finding and provide a more comprehensive analysis.

7. Several minor typographical errors are present in the paper. Correcting these errors is essential for improved readability and credibility.

8. The axes labels and tick labels in the figures are too small, affecting clarity. Enhancing the font sizes of these labels would significantly improve readability.

9. The conclusion section is not thoroughly presented.

Reviewer #3: Hello, Author. Thank you for submitting this manuscript for review. I want to commend the thoroughness of the research; however, I have highlighted its strengths and some areas where it could be improved.

1. This study addresses relevant practical problems like long flowback in tight reservoirs. It employs suitable experimental methods such as coreflooding and NMR T2 analysis to examine the displacement efficiency of different fracturing fluid gel-breakers. The experimental design controls key variables such as permeability through core pairs, interfacial tension, and viscosity to isolate their effects. Using NMR for pore-level displacement measurement is a strength. The conclusions about pore size impact, viscosity, interfacial tension, and time dependency are reasonable extensions of the experimental data.

Tables 2, 3, and 4 and Figures 4, 5, and 6 generally support the main conclusions regarding the effects of viscosity (Table 2), surfactants/interfacial tension (Table 3), and displacement time (Table 4) on overall and pore-specific displacement efficiency. The data support the observation of negative efficiency in micropores and plausibly explained.

However, the main weakness is the lack of replication. The study uses only one core per condition (e.g., one core for guanidine gum, one for cross-linked gum, one for slickwater, one for slickwater+surfactant). Although cores from longer samples were used for some comparisons (cores 1-2, 3-4), this does not constitute true replication. Natural heterogeneity within a single long core means results from one sample may not be representative. Without replicates, the reliability and general applicability of the efficiency values (e.g., 23.25% vs 18.31%, 13.11% vs 16.73%) are significantly reduced. While the directions of effects (e.g., lower viscosity increases efficiency) are probably valid due to controlled comparisons, the effect magnitudes and absolute efficiency values lack solid statistical support since n=1 per condition. The time-dependent study (Core 5) also involved only one core sequentially.

2. The manuscript reports no statistical analysis at all. Results are presented as single measurements (e.g., displacement efficiencies in tables) without any variance measures (standard deviation, error bars), significance tests (t-tests, ANOVA), or discussion of uncertainty. Given that only one core per condition was used (n=1), rigorous statistical comparison of treatments is impossible. The lack of acknowledgment of this limitation or attempts to quantify uncertainty (such as repeated measurements if feasible) means the analysis is not rigorous or appropriate.

3. The manuscript does not specify that raw data (like NMR T2 spectra, core properties beyond permeability, fluid measurements) have been deposited in a public repository or provided as supplementary material. While summary results are shown in figures and tables, raw data necessary for reproduction or re-analysis are not explicitly available in the manuscript or declared as accessible elsewhere.

4. The manuscript is generally clear and follows the conventional structure (Abstract, Intro, Methods, Results, Discussion, Conclusion). Technical terminology is appropriate. However, there are some instances of non-standard English and grammatical errors that slightly hinder readability but do not obscure the meaning:

a."There usually have a long backflow period" (Abstract - should be "is").

b."produced by the well soak" (Abstract - phrasing is awkward).

c."reservoir reservoirs" (Abstract - redundant)

d.Inconsistent capitalization (e.g., "Guanidine gum" vs "guanidine gum").

e.Sentences are often lengthy and would benefit from concise rewriting and professional copyediting.

**Do you want your identity to be public for this peer review?** For information about this choice, including consent withdrawal, please see our Privacy Policy

Reviewer #1: No

Reviewer #2: No

Reviewer #3: **Yes: ** Adeniyi Adebayo

---

## [Author Response · Author response to Decision Letter 1]

16 Sep 2025

Re: The Itemized Response to Comments Made by Editor

Journal: PLOS One

Manuscript ID: PONE-D-25-35735

Title: Study on crude oil displacement efficiency by fracturing fluid in tight sandstone reservoir

Dear Academic Editor:

Thank you for your letter and for the comments and requirements concerning our manuscript entitled “Study on crude oil displacement efficiency by fracturing fluid in tight sandstone reservoir” (ID: PONE-D-25-35735). Those comments and requirements are all valuable and very helpful for revising and improving our paper, as well as the important guiding significance to our researches. We have studied comments and requirements carefully and have made correction which we hope meet with approval. Revised portion are marked in red in the paper. The main corrections in the paper and the responds to the comments and requirements are as flowing:

Comment #1: Please ensure that your manuscript meets PLOS ONE's style requirements, including those for file naming. The PLOS ONE style templates can be found at https://journals.plos.org/plosone/s/file?id=wjVg/PLOSOne formatting sample main body.pdf and https://journals.plos.org/plosone/s/file?id=ba62/PLOSOne formatting sample title authors affiliations.pdf.

Response: Thank you very much for your reminder regarding the formatting requirements of PLOS ONE. We have carefully revised the manuscript according to the journal’s style guidelines, including file naming conventions, title and author formatting, and manuscript structure.

Action: All formatting has been adjusted to meet PLOS ONE’s requirements. The detailed modifications can be found in the revised manuscript (highlighted in red).

Comment #2: We note that this submission includes NMR spectroscopy data. We would recommend that you include the following information in your methods section or as Supporting Information files:

1) The make/source of the NMR instrument used in your study, as well as the magnetic field strength. For each individual experiment, please also list: the nucleus being measured; the sample concentration; the solvent in which the sample is dissolved and if solvent signal suppression was used; the reference standard and the temperature.

2) A list of the chemical shifts for all compounds characterized by NMR spectroscopy, specifying, where relevant: the chemical shift (δ), the multiplicity and the coupling constants (in Hz), for the appropriate nuclei used for assignment.

3)The full integrated NMR spectrum, clearly labelled with the compound name and chemical structure.

We also strongly encourage authors to provide primary NMR data files, in particular for new compounds which have not been characterized in the existing literature. Authors should provide the acquisition data, FID files and processing parameters for each experiment, clearly labelled with the compound name and identifier, as well as a structure file for each provided dataset. See our list of recommended repositories here: https://journals.plos.org/plosone/s/recommended-repositories.

Response: Thank you very much for your detailed comment on the NMR spectroscopy data. We would like to clarify that the NMR tests conducted in this study are different from conventional NMR spectroscopy used for compound characterization. Specifically, our experiments employed low-field Nuclear Magnetic Resonance (NMR) technology, which is widely used to characterize the distribution of fluids in porous media. The purpose of these tests was to investigate the pore-scale oil displacement characteristics of tight sandstone cores.

In our study, the principle of low-field NMR is as follows: under an external magnetic field, the hydrogen nuclei of crude oil resonate when their oscillation frequency matches the applied frequency, absorbing energy and generating an NMR signal. After the radiofrequency pulse is turned off, the nuclei release energy and gradually return to equilibrium during a relaxation process. The echo attenuation signals collected from oil-saturated cores are then processed using mathematical inversion methods to obtain T2 spectra, which provide information about the pore structure and fluid distribution within the cores. According to the journal’s requirements, we have supplemented the detailed information of the NMR instrument (make, source, and magnetic field strength) in the Experimental Section. Additionally, we have provided in the Supporting Information files the detailed principles of the NMR characterization and the original NMR data files of the tested core samples.

Action: The following statement was added to Experimental Section:

“In this study, a MesoMR23-060 NMR imaging system (Suzhou Niumag Analytical Instrument Corporation, Suzhou, China) was used for core testing to investigate the distribution of crude oil within the core samples. This equipment is equipped with an independent pulse control module and a radio frequency (RF) transmission/reception circuit, which improves the accuracy and stability of pulse sequences and enhances the signal-to-noise ratio of NMR signals. The spectrometer frequency of the RF pulse used in this study was 21 MHz, and the waiting time between two scans of sampling data was 4000 ms.”

And the basic principles of NMR tests for characterizing the distribution of crude oil within the cores, as well as the original NMR data, are provided in the Supporting Information.

Comment #3: We note that your Data Availability Statement is currently as follows: All relevant data are within the manuscript and its Supporting Information files.

Response: We sincerely thank the editor for this helpful reminder. We confirm that all raw data required to replicate the results of this study have been fully provided. The detailed datasets, including the data used to construct the figures, as well as the original NMR data, are all included in the Supporting Information files. These materials ensure that the minimal data set defined by PLOS ONE is openly available for replication and verification.

We greatly appreciate the editor’s careful consideration and constructive guidance.

Action: All the original data underlying this study are provided in the Supporting Information files, where the details can be fully accessed.

Comment #4: Thank you for stating the following financial disclosure:

This research is supported by the National Natural Science Foundation of China (No.52174031) and the Youth Innovation Team of Shaanxi Universities.

Response: We sincerely thank the editor for the reminder regarding the financial disclosure. We confirm that the funders had no role in study design, data collection and analysis, decision to publish, or preparation of the manuscript.

Comment #5: Thank you for stating the following in the Acknowledgments Section of your manuscript: This research is supported by the National Natural Science Foundation of China (No.52174031) and the Youth Innovation Team of Shaanxi Universities.

This research is supported by the National Natural Science Foundation of China (No.52174031) and the Youth Innovation Team of Shaanxi Universities.

Response: We sincerely thank the editor for the reminder regarding the placement of funding information. We have carefully revised the manuscript and removed all funding-related text from the Acknowledgments section and other parts of the manuscript.

The Funding Statement provided by the editor - “This research is supported by the National Natural Science Foundation of China (No.52174031) and the Youth Innovation Team of Shaanxi Universities.” - is correct. We have also included this Funding Statement in our cover letter as requested.

We greatly appreciate the editor’s kind guidance and support.

Action: All funding-related text has been deleted from the manuscript, and the correct Funding Statement has been included in the cover letter.

Comment #6: Please amend your authorship list in your manuscript file to include author Jinsheng Zhao.

Response: We sincerely thank the editor for this reminder. We have carefully revised the manuscript and updated the authorship list to include Jinsheng Zhao as one of the co-authors.

Action: The authorship list in the revised manuscript file has been amended accordingly.

Comment #7: Please amend the manuscript submission data (via Edit Submission) to include author Jinfeng Li.

Response: We sincerely thank the editor for this helpful reminder. We have carefully revised the manuscript submission information in the online system and updated it to include Jinfeng Li as one of the co-authors.

Action: The submission data have been amended accordingly to include Jinfeng Li.

Comment #8: If the reviewer comments include a recommendation to cite specific previously published works, please review and evaluate these publications to determine whether they are relevant and should be cited. There is no requirement to cite these works unless the editor has indicated otherwise.

Response: We sincerely thank the editor for this reminder. We have carefully reviewed and evaluated the references suggested by the reviewers. Based on their relevance and contribution to our study, we have made appropriate judgments and incorporated citations where they benefit the revision of our manuscript.

We greatly appreciate the constructive suggestions and guidance from the reviewers and the editor.

Re: The Itemized Response to Comments Made by Reviewer #1

Journal: PLOS One

Manuscript ID: PONE-D-25-35735

Title: Study on crude oil displacement efficiency by fracturing fluid in tight sandstone reservoir

Dear Editors and Reviewers:

Thank you for your letter and for the comments concerning our manuscript entitled “Study on crude oil displacement efficiency by fracturing fluid in tight sandstone reservoir” (ID: PONE-D-25-35735). Those comments are all valuable and very helpful for revising and improving our paper, as well as the important guiding significance to our researches. We have studied comments carefully and have made correction which we hope meet with approval. Revised portion are marked in red in the paper. The main corrections in the paper and the responds to the reviewer’s comments are as flowing:

This manuscript focuses on the study of the displacement efficiency of fracturing fluid on crude oil in tight sandstone reservoirs, conducting relevant experiments, and exhibiting certain innovativeness and practicality. However, there are some deficiencies need to be checked out and revised:

Comment #1: In Introduction section, author mentioned imbibition, but there is a lack of the relevance of imbibition to the main work of this manuscript. It is recommended that author should clarify this relevance to enhanced the readability of the manuscript.

Response: We sincerely thank the reviewer for this constructive suggestion. We agree that it is important to clarify the role of imbibition in relation to the main work of this manuscript. In the revised Introduction, we have added specific explanations to highlight that the displacement efficiency of fracturing fluids is primarily determined by the imbibition efficiency during the soaking stage. Specifically, during the soaking stage, the fracturing fluid spontaneously enters into the rock pores through the imbibition effect, displacing the crude oil within them. This process usually occurs between the fractures created by fracturing and the reservoir matrix. After the soaking stage, the displaced crude oil is produced together with the fracturing fluid during the flowback stage. In addition, we have supplemented the background description of imbibition in the Introduction to further clarify its importance in tight reservoirs. This addition strengthens the logical connection between the imbibition mechanism and the objectives of this study, demonstrating that imbibition plays a central role in evaluating the oil displacement efficiency of different fracturing fluids.

Action: The following statement was added to Introduction:

“During the soaking stage, the fracturing fluid spontaneously enters into the rock pores through the imbibition effect, displacing the crude oil within them. This process usually occurs between the fractures created by fracturing and the reservoir matrix. After the soaking stage, the displaced crude oil is produced together with the fracturing fluid during the flowback stage, forming an oil-water mixture that significantly increases crude oil production and reduces the flowback volume of fracturing fluid.”

“Imbibition is more common in low-permeability and tight reservoirs due to the presence of microscopic pore throats and fractures formed during hydraulic fracturing. Since the 1950s, imbibition has been extensively studied and developed as an important method of oil production.”

Comment #2: The relationship of imbibition and the flowback of fracturing fluid should be clarified in introduction (for example, the flowback of fracturing fluid may be achieved through imbibition).

Response: We thank the reviewer for this insightful comment. We agree that the relationship between imbibition and the flowback of fracturing fluid should be clarified in the Introduction. In the revised manuscript, we have highlighted that the soaking stage involves spontaneous imbibition of fracturing fluid into the rock pores, displacing crude oil that is subsequently produced during flowback. Furthermore, the flowback of fracturing fluid in tight reservoirs is closely related to the imbibition process, as imbibition not only displaces crude oil but also facilitates the return of frac

---

## [Decision Letter · Decision Letter 1]

8 Oct 2025

Study on crude oil displacement efficiency by fracturing fluid in tight sandstone reservoir

PONE-D-25-35735R1

Dear Dr. zhao,

We’re pleased to inform you that your manuscript has been judged scientifically suitable for publication and will be formally accepted for publication once it meets all outstanding technical requirements.

Kind regards,

Lisong Zhang

Academic Editor

PLOS ONE

Additional Editor Comments (optional):

Reviewers' comments:

Reviewer's Responses to Questions

**Comments to the Author**

Reviewer #1: All comments have been addressed

Reviewer #3: All comments have been addressed

2. Is the manuscript technically sound, and do the data support the conclusions?

Reviewer #1: Yes

Reviewer #3: Yes

3. Has the statistical analysis been performed appropriately and rigorously?

Reviewer #1: Yes

Reviewer #3: Yes

4. Have the authors made all data underlying the findings in their manuscript fully available?

Reviewer #1: Yes

Reviewer #3: Yes

5. Is the manuscript presented in an intelligible fashion and written in standard English?

Reviewer #1: Yes

Reviewer #3: Yes

Reviewer #1: (No Response)

Reviewer #3: Hello Author, I am pleased with your responses and the actions taken to address the issues I raised in the first submission.

**Do you want your identity to be public for this peer review?** For information about this choice, including consent withdrawal, please see our Privacy Policy

Reviewer #1: No

Reviewer #3: No

---

## [Editor Report · Acceptance letter]

PONE-D-25-35735R1

PLOS ONE

Dear Dr. zhao,

I'm pleased to inform you that your manuscript has been deemed suitable for publication in PLOS ONE. Congratulations! Your manuscript is now being handed over to our production team.

Kind regards,

on behalf of

Associate Professor Lisong Zhang

Academic Editor

PLOS ONE